# Therapeutic Potential of Targeting the HMGB1/RAGE Axis in Inflammatory Diseases

**DOI:** 10.3390/molecules27217311

**Published:** 2022-10-27

**Authors:** Harbinder Singh, Devendra K. Agrawal

**Affiliations:** Department of Translational Research, College of Osteopathic Medicine of the Pacific, Western University of Health Sciences, Pomona, CA 91766, USA

**Keywords:** HMGB1, RAGE, inflammation, inhibitors, TLRs

## Abstract

High mobility group box 1 (HMGB1) is a nuclear protein that can interact with a receptor for advanced glycation end-products (RAGE; a multi-ligand immunoglobulin receptor) and mediates the inflammatory pathways that lead to various pathological conditions, such as cancer, diabetes, neurodegenerative disorders, and cardiovascular diseases. Blocking the HMGB1/RAGE axis could be an effective therapeutic approach to treat these inflammatory conditions, which has been successfully employed by various research groups recently. In this article, we critically review the structural insights and functional mechanism of HMGB1 and RAGE to mediate inflammatory processes. More importantly, current perspectives of recent therapeutic approaches utilized to inhibit the communication between HMGB1 and RAGE using small molecules are also summarized along with their clinical progression to treat various inflammatory disorders. Encouraging results are reported by investigators focusing on HMGB1/RAGE signaling leading to the identification of compounds that could be useful in further clinical studies. We highlight the current gaps in our knowledge and future directions for the therapeutic potential of targeting key molecules in HMGB1/RAGE signaling in the pathophysiology of inflammatory diseases.

## 1. Introduction

High mobility group box 1 (HMGB1) is a chromatin-binding nuclear protein that binds to DNA and regulates various transcription factors such as NF-κB and glucocorticoid receptors [1,2]. Its excessive amount in extracellular space can cause cellular or tissue injury and organ dysfunction leading to various pathological conditions [3,4,5,6,7]. HMGB1 is a family of three nuclear proteins in mammals consisting of HMGB1, HMGB2, and HMGB3 [7,8]. HMGB1 was first discovered 45 years ago as a non-histone chromatin-based protein having electrophoretic mobility [9]. HMGB1 is expressed in almost all eukaryotic cells and is involved in the maintenance and regulation of cellular structure, gene transcription, and transcriptional factors [10,11]. Its conserved form has several biological functions inside as well as outside of the cell. Intracellularly, when translocated to the cytoplasm, it can intervene in autophagy. Similarly, inside the nucleus, it can strongly interact with histone and DNA to regulate the process of transcription by determining the chromatin structure [4]. Outside the cell, HMGB1 serves as an alarmin molecule, also known as damage-associated molecular pattern (DAMP) [2]. Extracellularly, it has the ability to interact with several molecules such as pathogen-associated molecular patterns (PAMPs), cytokines, and chemokines [2].

Structurally, HMGB1 is a protein of 29 kDa consisting of 215 amino acids having two N-terminal DNA binding domains, viz., HMG box-A and HMG box-B, having 9–79 and 89-163 amino acids sequences, respectively (Figure 1). The third domain is a C-terminal acidic tail that consists of a 186–215 amino acid sequence (Figure 1). The amino acids of N- and C-terminal regions are basic and acidic in nature, respectively, and can exert their diverse functions upon binding with various factors [12]. The basic amino acids containing boxes have the ability to bind with DNA and help in bending and distortion of the double helix [13], while the acidic region (C-terminus) helps to regulate the DNA-binding specificity of HMGB1 [14]. HMGB1 also has the ability to bind with distorted forms of DNA, such as UV- and cisplatin-damaged DNA and cruciform DNA [15]. Therefore, this protein was identified as a key participating factor in the nuclear events of DNA recombination, replication, remodeling, and repair [16,17].

Extracellular HMGB1 can bind with the receptor for advanced glycation end products (RAGE), toll-like receptors (TLRs), and thus, can mediate inflammation [18,19]. RAGE is a 45–50 kDa multi-ligand transmembrane receptor consisting of three extracellular immunoglobulin-like domains: First, the extracellular domain with 23–342 amino acid residues; second is hydrophobic transmembrane domain with 343–363 amino acid residues; and third, is an intracellular cytoplasmic domain with 404–464 residues (Figure 1) [20]. The extracellular region is further subdivided into three immunoglobulin-like domains: Variable domain (V-domain) having 23–116 amino acid residues which is connected to two constant domains C1 (residues 124–221) and C2 (residues 227–317) (Figure 1) [21]. Its VC1 domain provides the major contribution towards the interaction with the vast number of ligands that can stimulate various pathological inflammatory pathways (Figure 1 and Figure 2). The overall mechanisms of the HMGB1 ligand binding with RAGE to overexpress the inflammatory cytokines such as tumor necrosis factor (TNF-α) and interleukins (IL-6 and IL-1β) are depicted in Figure 2. The detailed functional description of individual domains of RAGE was recently described in our publication [22].

Besides the RAGE, HMGB1 also have the ability to interact with toll-like receptors (TLR2/TLR4). There are many properties that have been found to regulate TLR signaling. Numerous evidence support the direct crosstalk between RAGE ligands and TLRs. The binding ability of HMGB1 is controlled by its various states [23]. Specifically, its redox state with two cysteine residues (Cys23 and Cys45) is mainly responsible for showing the interaction with TLR4 [23]. Mutations on these two specific amino acid residues lowers the affinity of HMGB1 towards TLR4 as well as its complex with myeloid differentiation factor-2 (MD2) [23]. It has been confirmed with surface plasmon resonance (SPR) that its reduced form binds to the TLR4/MD2 complex with approximately 10-fold lower affinity and can bind to MD2 individually with 100-fold lower affinity than its oxidized form [24]. This crosstalk was also confirmed through ex vivo experiments. The exposure of isolated mouse spleen cells to HMGB1 enhances cytokine excretion and was found to be initiated through TLR2 or TLR4 compared to the cells not previously treated with HMGB1 [25]. Similarly, HMGB1 was found in bone crush mixture and the exposure of macrophages to this mixture resulted in the upregulation of caveolin-1 expression in a RAGE-dependent fashion, which, in turn, induced caveolae-mediated TLR4 internalization and desensitization of macrophages [25]. Bone-marrow derived macrophages (BMDM) produced very low levels of cytokines (TNF-α, IL-1β, and IL-6) in RAGE-deficient mice compared to the wild-type BMDM following HMGB1 treatment [26]. This suggests that RAGE is physically interacting with the TLRs to elicit various inflammatory functions.

## 2. Expression of HMGB1/RAGE and Their Biological Functions

In 1999, Wang et al. [27] reported that HMGB1 is an important extracellular mediator for inflammatory processes. This protein is released in active and passive states from the dendritic macrophagic cells and from necrotic cells, respectively, and is associated with various pathological conditions (Figure 2) [27]. Post translational modifications such as acetylation, methylation, phosphorylation, and oxidation generally induce the active secretion of HMGB1, which is mediated through lysosomes [28], although, the exact mechanism of its active secretion is still unclear; however, it is revealed that upon infection, C5a engages with its receptor C5aR2 in macrophages that upregulates the expression of HMGB1 and is secreted through intracellular signaling [29].

The expression of RAGE is constitutively high in lungs and skin cells throughout life. However, its expression in vascular smooth muscle cells, endothelial cells, monocytes and macrophages, neutrophils, and neurons is very low under physiological conditions. Accumulation of AGEs or response to the inflammatory mediators increases RAGE expression [30,31,32,33,34,35,36]. Similarly, HMGB1 binding to RAGE stimulates distinct signaling molecules such as NF-κB, p38, kinases such as ERK1/2 IRAK, cdc42, etc. [37] (Figure 2). Hori et al. first reported that the amino acids of box-A (23–50), amino acids of box-B, and acidic tail (150–183) of HMGB1 are primarily responsible for its binding with RAGE (Figure 1) and thus activate these inflammatory signaling pathways and mediate several disease conditions [38,39].

**Figure 2 molecules-27-07311-f002:**
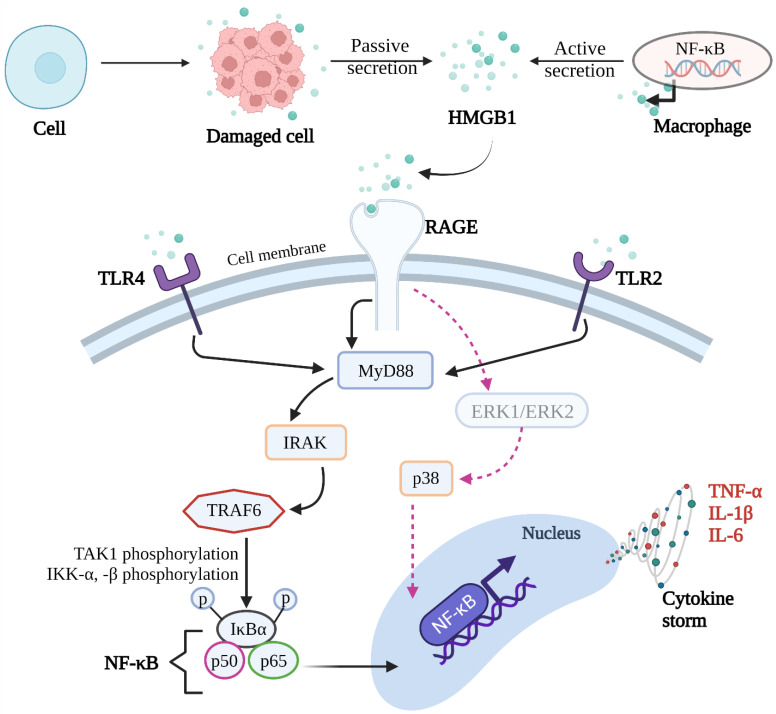
Active and passive secretion of HMGB1 and its interactions with RAGE, TLR2, and TLR4 to regulate the inflammatory cascade.

Considering the role of HMGB1 and RAGE in various pathological conditions, it has been demonstrated that blocking their mutual interactions represents a promising strategy to regulate the inflammation associated with various disease conditions. In this article we reviewed several recent approaches reported in the literature to inhibit HMGB1 based on blocking the HMGB1/RAGE axis.

## 3. Molecules Targeting HMGB1/RAGE Axis in Various Inflammatory Diseases

### 3.1. Crocin

It is known that glucagon-like peptide-1 (GLP-1) has the ability to attenuate the expression of HMGB1. In this instance, Tabaa et al. [40] checked the potential of GLP-1 stimulator, i.e., crocin (a carotenoid chemical structure; Figure 3) against HMGB1 in the treatment of cigarette smoking-induced cognitive impairments [40]. A rat model was used for the evaluation of crocin, and expression of GLP-1, HMGB1, and other pro-inflammatory markers in the hippocampus were determined by using ELISA, Western blotting, and immunohistochemistry. Initially, cognitive functions were tested using a Morris water maze (MWM), elevated plus maze (EPM), and passive avoidance methods. The cognitive impairment was induced by using cigarette smoke exposure and treatment with GLP-1 stimulator considerably decreased HMGB1 in cigarette smoke exposed rats. With three weeks of treatment with 30 mg/kg of crocin, GLP-1 level was upregulated in the rat hippocampus. The significant decrease in the level of HMGB1 and RAGE was observed in crocin-treated rat hippocampus compared to control rats. Downstream pro-inflammatory markers were further tested using immunohistochemistry and Western blotting which were also significantly attenuated in crocin-treated rat hippocampal tissues. These findings suggest the neuroprotective effect of crocin via suppressing the HMGB1/RAGE axis [40].

### 3.2. Berberine

For the treatment of sepsis-associated encephalopathy (SAE), Shi et al. [41] used berberine (Figure 3) and claimed that this treatment can considerably ameliorate the memory impairment in sepsis mice. The berberine treatment of mice significantly reduced the level of pro-inflammatory cytokines (TNF-α, IL-1α) in the hippocampus and also showed neuroprotective effect. To evaluate the molecular mechanism, the binding of berberine with HMGB1 was analyzed using molecular docking studies, which revealed that the berberine can strongly bind to HMGB1 with the dock score of −7.69. The upregulated HMGB1-induced TNF-α was also attenuated with the berberine treatment, which was examined by employing in vitro method using microglia and astrocytes. Further, to confirm that the berberine attenuates the downstream signaling through the HMGB1/RAGE axis, a *RAGE*^−/−^ knock out mice model was used to examine the effect of berberine and it was found that berberine did not improve the cognitive impairment in the *RAGE*^−/−^ mice, suggesting the inhibitory effect of berberine specifically via the HMGB1/RAGE axis [41].

### 3.3. FPS-ZM1 and Adriamycin

Lai et al. [42] reported that the interaction of HMGB1 with RAGE upregulates autophagy in the treatment of acute leukemia and develops resistance against its treatment. To abort these interactions, they used Adriamycin, also known as doxorubicin, (to inhibit HMGB1) and FPS-ZM1 (to inhibit RAGE) (Figure 3) in leukemia cells and checked the expression of HMGB1 and RAGE using Western blotting. Adriamycin upregulated the apoptosis of leukemic cells in a dose-dependent manner and at higher concentration (0.4 μM) the level of HMGB1 significantly decreased. RAGE expression was also significantly decreased with the treatment with FPS-ZM1 [42].

### 3.4. Curcumin

Similarly, Han et al. [43] also reported the potential of curcumin (Figure 3) to treat the cognitive impairment in a transgenic mice model claiming the possibility of inhibiting the HMGB1/RAGE inflammatory pathway [43]. For the evaluation of curcumin, they used APP/PS1 transgenic mice for the diseased group, and wild type (WT) mice as the control. The curcumin diet of 100 mg/kg/d was provided to the mice in the treatment group for five months and was started at the age of four months and memory function was determined using a Morris water maze (MWM) test and Y-maze model. Curcumin-treated groups of mice exhibited improvement in their learning and memory performance, which was based on the decreased escape latency time. After evaluating memory functions, the brains of these mice were isolated to investigate the inflammatory signaling pathway to understand the mechanism of action of curcumin. The protein expression of HMGB1, RAGE, TLR4, and NF-κB was examined by Western blotting. The expression of all these proteins was significantly increased in the hippocampus of the diseased mice (APP/PS1 mice) compared to WT. It is noteworthy that curcumin did not affect the plaque load in the hippocampus of the transgenic mice suggesting the possible mechanism of curcumin to show neuroprotection specifically through the HMGB1/RAGE inflammatory pathway [43].

### 3.5. Glycyrrhizin & Pentoxifylline

Glycyrrhizin has been found to be the positive binder of HMGB1 that can inhibit the interactions of HMGB1 to RAGE to further block the inflammatory pathway [44]. In this regard, Okuma et al. [45], in 2014, evaluated the potential of glycyrrhizin (Figure 3) to treat traumatic brain injury using the fluid percussion-induced injury rat model. Initially, the rotarod apparatus was used to access the motor function in rats which was performed at 3, 6, and 24 h after brain injury. The results revealed the dose-dependent effect of glycyrrhizin (Figure 3) on coordinated motor activity and the walking time period was significantly increased compared to vehicle-treated and diseased rats. Rat brain tissues were isolated to determine the protein and mRNA expression of HMGB1 and RAGE with the treatment of glycyrrhizin. Intravenous administration of 4 mg/kg of glycyrrhizin considerably decreased the translocation of HMGB1 in neuronal cells and maintained their activity. The plasma level of HMGB1 was also decreased in animals treated with 4 mg/kg of glycyrrhizin compared to vehicle control animals. Similarly, treatment with glycyrrhizin significantly decreased the mRNA transcripts of pro-inflammatory cytokines (TNF-α, IL-1β, and IL-6). To examine if the neuroprotective effect of glycyrrhizin works specifically through the HMGB1/RAGE axis, its effect was examined in *RAGE*^−/−^ knock-out mice. Mice treated with 4 mg/kg of glycyrrhizin did not produce any inhibitory effect suggesting that glycyrrhizin specifically affects the HMGB1/RAGE pathway to improve the neuro-inflammatory condition [45]. The hemorheological agent pentoxifylline (Figure 3) also showed its anti-epileptic effect through HMGB1/RAGE axis [46].

### 3.6. Dexmedetomidine

Very recently, the potential of dexmedetomidine (Figure 4) to act through the HMGB1/RAGE axis was also identified [47]. For the treatment of acute lung injury (ALI) they evaluated the effect of dexmedetomidine using in vitro as well as in vivo methods. In the in vitro method, the MLE-12 cells (mouse lung type II epithelial cell line) were treated with lipopolysaccharide (LPS) and in the in vivo model the cecal ligation perforation (CLP) stimulated ALI mice model was used. After inducing the injury in the treatment and control mice groups, protein and genetic expression levels of HMGB1, RAGE, and NF-κB were examined using Western blot analysis and qRT-PCR, respectively. Hematoxylin and eosin staining was also done to check the tissue injury. Lung tissue collected from the dexmedetomidine (10 μg/kg i.p.)-treated mice showed that its treatment significantly attenuated lung tissue damage. Genetic and protein expression was also significantly decreased in the treatment groups. Moreover, cellular studies with MLE-12 cells revealed that dexmedetomidine (Figure 4) treatment significantly suppressed HMGB1 translocation from the nucleus to the cytoplasm, and this effect was reversed by RAGE overexpression. Therefore, dexmedetomidine could be considered as an effective treatment option for the ALI which acts through suppressing the HMGB1/RAGE pathway [47].

### 3.7. Ketamine

In 2018, Zhang et al. [48] proposed a new molecular mechanism of ketamine (a known anesthetic compound; Figure 4) to treat ALI and proposed that it blocks the HMGB1/RAGE signaling pathway. To confirm this, they used an LPS-induced ALI male Wistar rat model in which animals were injected with 10 mg/kg of LPS through their femoral veins. Blood gas analysis was performed to check the arterial blood oxygen partial pressure and pH, and compared between the control and 50 mg/kg ketamine (injected through femoral vein)-treated animal groups. With the treatment with ketamine, the partial pressure of arterial oxygen and pH were significantly increased, which were decreased in the LPS-induced ALI animals. Lung tissues were collected from the control and ketamine (Figure 4)-treated animals to access the protein and mRNA expression of HMGB1 and RAGE using rt-PCR and Western blotting, respectively, to check whether ketamine could act through the HMGB1/RAGE axis or not. Significant reduction was found in both the protein and mRNA expression in the tissues collected from ketamine-treated rats compared to the LPS-treated rats, which supports the action of ketamine to block the HMGB1/RAGE signaling pathway to treat ALI in rats [48]. This study corroborated the previous findings of Li et al. [49], which also suggested the potential effect of ketamine acting through HMGB1/RAGE axis to treat sepsis-induced ALI.

### 3.8. Epigallocatechin-3-Gallate

Treatment with (−)-epigallocatechin-3-gallate (EGCG: most abundant constituent of green tea; Figure 4) has the ability to inhibit osteoclast differentiation, however the exact mechanism of action is unknown. Recently, in 2020, Nishioku et al. [50] reported various potential mechanisms of action of EGCG amongst which HMGB1/RAGE is the one through which the inflammatory condition can be treated. They found that the extracellular release of the HMGB1 was significantly decreased with the treatment with EGCG, which was determined by using Western blotting analysis in cell lysates of osteoclast precursor cells isolated from the bone marrow of femora and tibiae of C57BL/6N mice [51]. The protein expression of RAGE was also significantly decreased with the treatment of EGCG. Therefore, the study claimed the potential of EGCG to reduce the bone-resorbing activity of osteoclasts acting through the HMGB1/RAGE pathway [50].

### 3.9. Telmisartan, Irbesartan and Candesartan

There is a well-established literature related to the sartan compounds such as telmisartan, irbesartan, and candesartan (Figure 4) suggesting their role in treating various inflammatory conditions associated with cardiovascular diseases via the HMGB1/RAGE axis [52]. The use of the drug molecules proved to be efficacious in the prevention and acute treatment of stroke, reducing RAGE expression, thus inhibiting the HMGB1/RAGE axis in stroke conditions [52].

## 4. Anti-Inflammatory Effect of HMGB1/RAGE Axis

Besides the role of the HMGB1/RAGE axis in inflammatory diseases, several molecules can also switch their pro-inflammatory action to anti-inflammatory functions. This could regulate the immune tolerance by initiating the release of anti-inflammatory macrophages [53]. RAGE could perform various functions interacting through various membranous receptors, which led to exerting tolerogenic functions at the HMGB1/RAGE axis. Recent studies revealed that HMGB1 can promote leukotriene production that induces TRAF5 in a RAGE-dependent manner, and excretes pro-resolving mediators (SPMs), which act as anti-inflammatory mediators [53,54,55]. This suggests that RAGE could not only facilitate inflammation but also has an anti-inflammatory effect. However, the investigation of the immune homeostasis regulating the role of RAGE in various conditions is warranted. Further studies are required to determine the environmental factors that lead to the inflammatory as well as anti-inflammatory function of the HMGB1/RAGE axis.

## 5. Conclusions and Future Directions

In the past two decades, HMGB1 and RAGE are the well-characterized therapeutic targets for inflammatory diseases. The unique feature of these molecules is that they operate in opposite directions in alarming conditions, such as chronic inflammation. The extracellular secretion of HMGB1 mediates the inflammatory response upon binding with RAGE, thus activating the release of pro-inflammatory cytokines such as TNF-α, IL-6, and IL-1β through NF-κB signaling pathways.

A number of approaches have been proposed to block the HMGB1/RAGE axis to treat inflammatory diseases such as Alzheimer’s disease, ALI, cardiovascular diseases, osteoarthritis, and others. Several strategies utilized small molecules that directly and efficiently interact with HMGB1 or RAGE and some of them act as competitive antagonists, for example, glycyrrhizin, curcumin, berberine and FPS-ZM1.

The most studied small molecule inhibitors of HMGB1/RAGE axis are glycyrrhizin and FPS-ZM1. Glycyrrhizin is a natural triterpene glycosidic compound that has been widely investigated in various HMGB1/RAGE-mediated diseases that confirm the inhibition of the extracellular HMGB1 cytokine effect in animal models. FPS-ZM1 is a small molecule that is a specific inhibitor of RAGE that further blocks the interactions of various RAGE ligands including HMGB1 and has thus far been tested in the treatment of various inflammatory conditions in animal models. Despite these studies, literature still lacks the clinical data involving the interface of the HMGB1/RAGE axis, which may be due to the lack of specific knowledge of the complex inflammatory signaling system. Therefore, it is critical to analyze various structural and biological features associated with the HMGB1/RAGE axis to block the interactions in order to develop new and specific therapeutics through various drug designing approaches.

One limitation of this compilation is that, despite the potential targets of the HMGB1/RAGE axis in various inflammatory diseases, other pathways are also involved in the pathophysiology of the diseases. For example, in neuroinflammatory diseases such as Alzheimer’s disease (AD), *β*-amyloid aggregation and cholinesterases are also the significant cause. In this article, the role of these targets including cyclooxygenases, lipoxygenases, reactive oxygen species, S100 proteins, *β*-amyloid, etc. and their inhibitors are not discussed.

## Figures and Tables

**Figure 1 molecules-27-07311-f001:**
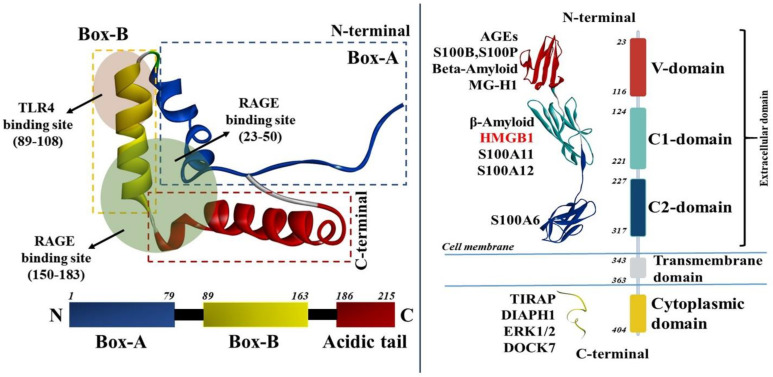
(**left**) Structure of HMGB1 and various ligand binding domains (PDB: 1HME); (**right**) Structure of full-length RAGE and various ligands that can bind to their respective binding domains (PDBs: 4LP5, 6VXG).

**Figure 3 molecules-27-07311-f003:**
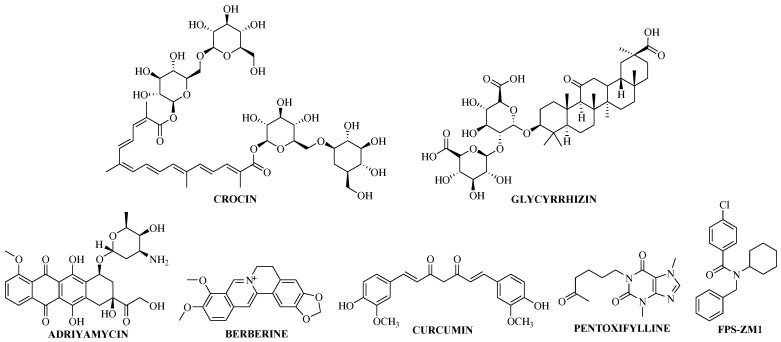
Molecules useful for the treatment of neuro-degenerative disorders acting through the HMGB1/RAGE axis.

**Figure 4 molecules-27-07311-f004:**
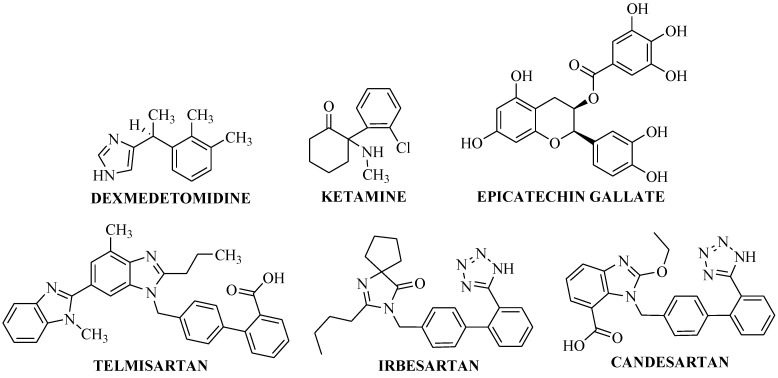
Molecules for the treatment of various inflammatory diseases acting via the HMGB1/RAGE axis.

## Data Availability

Relevant literature was downloaded from the online databases, including PubMed and Sciencedirect.com. Various combinations of keywords were used to find out the literature including ‘HMGB1 inhibitor’, ‘RAGE inhibitor,’ ‘HMGB1-RAGE’ and ‘HMGB1/RAGE axis’. After significant data collection, it was critically reviewed and summarized in systematic manner. For the structural analysis of proteins or receptors, PDB codes (1HME, 4LP5, and 6VXG) for crystallized structures were obtained from protein data bank (rcsb.org) and figures were generated using discovery studio and Microsoft-PowerPoint.

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
