# Peer review of "Therapeutic Potential of Targeting the HMGB1/RAGE Axis in Inflammatory Diseases"

_molecules, 2022, doi:10.3390/molecules27217311_

Round 1

Reviewer 1 Report

The manuscript focused on the HMGB1/RAGE axis’s potential therapeutic approach in inflammatory diseases.

It is an interesting subject. However, some issues must be addressed.

The sentence on line 92 is too strong and imprecise because the cited diseases have other components besides HMGB1/RAGE's role.

On topic 3, it would be better to put together the molecules by mechanisms of action because FPS-ZM1 and Curcumin, only 1 article was cited. Therefore,  futher explain the mechanisms of action.

Another point that should be better explored is the HMGB1 intracellular and extracellular effects.

A limitation section should be included explaining that despite the potential target of HMGB1/RAGE axis’s other pathways are also involved in the pathophysiological mechanism of discussed diseases.

Add a section discussing the crosstalking TLR 2 and 4 and RAGE.

Also, include potential anti-inflammatory effects of HMGB1/RAGE axis’s.

Author Response

Response to the concerns of Reviewer #1

  1. The sentence on line 92 is too strong and imprecise because the cited diseases have other components besides HMGB1/RAGE's role.

Author Response and Action Taken: The sentence has been rephrased (Page no. 6).

  1. On topic 3, it would be better to put together the molecules by mechanisms of action because FPS-ZM1 and Curcumin, only 1 article was cited. Therefore, further explain the mechanisms of action.

Author Response and Action Taken: Thank you for this suggestion. All the molecules have been described with their mechanism of action with respect to HMGB1/RAGE axis. The molecules have been individually described just because of the better readability. In our opinion, compiling all the molecules in a single paragraph would lose the interest of the readers of this article.

  1. Another point that should be better explored is the HMGB1 intracellular and extracellular effects.

Author Response and Action Taken: Intracellular and extracellular functions of HMGB1 have now been described with the appropriate citations on page 3.

  1. A limitation section should be included explaining that despite the potential target of HMGB1/RAGE axis’s other pathways are also involved in the pathophysiological mechanism of discussed diseases.

Author Response and Action Taken: Thank you for this suggestion. The limitation of the review article is included now in the conclusion and future perspective section.

  1. Add a section discussing the crosstalking TLR 2 and 4 and RAGE.

Author Response and Action Taken: The section describing the cross-talk between TLRs and RAGE has been discussed now in the manuscript on page 4 with appropriate citations.

  1. Also, include potential anti-inflammatory effects of HMGB1/RAGE axis’s.

Author Response and Action Taken: In the revised manuscript a new section ‘Anti-inflammatory effect of HMGB1/RAGE axis’ on page 12 has been included (Section 4).

Reviewer 2 Report

This review features discussions on the therapeutic potential of targeting the high mobility group box 1 (HMGB1)/ receptor for advanced glycation end products (RAGE) axis in inflammatory diseases. Extracellular HMGB1 can be bound with the RAGE and tall like receptors, thus mediating inflammation. The authors claim that blocking the HMGB1/RAGE axis could be an effective therapeutic approach to treat various inflammatory conditions. They introduced the role of HMGB1/RAGE and their biological functions and summarized molecules targeting HMGB1/RAGE axis in various inflammatory diseases. Therefore, the authors claimed that it is critical to analyze various structural and biological features associated with HMGB1/RAGE axis to develop new drugs. The following points should be clarified.

I recommend that the authors use tables and/or figures to summarize the molecules targeting HMGB1/RAGE axis in various inflammatory diseases.

Author Response

Response to the concerns of Reviewer #2

  1. I recommend that the authors use tables and/or figures to summarize the molecules targeting HMGB1/RAGE axis in various inflammatory diseases.

Author Response and Action Taken: Figure 3 and Figure 4 summarize the molecules targeting HMGB1/RAGE axis to treat various inflammatory diseases.

The authors sincerely appreciate the comments of the reviewers.  Revision of the manuscript according to the constructive comments of the reviewers significantly improved the quality and the information. We hope that the revised manuscript is now suitable for publication in 

Round 2

Reviewer 1 Report

The authors have addressed all issues raised.